# The TAS1R2 G-protein-coupled receptor is an ambient glucose sensor in skeletal muscle that regulates NAD homeostasis and mitochondrial capacity

Joan Serrano[1], Jordan Boyd[1], Ian S. Brown[1], Carter Mason[1], Kathleen R. Smith[1], Katalin Karolyi[1], Santosh K. Maurya[2], Nishita N. Meshram[1], Vanida Serna[1], Grace M. Link[1], Stephen J. Gardell[3] & George A. Kyriazis [1] ✉

The bioavailability of nicotinamide adenine dinucleotide (NAD) is vital for skeletal muscle health, yet the mechanisms or signals regulating NAD homeostasis remain unclear. Here, we uncover a pathway connecting peripheral glucose sensing to the modulation of muscle NAD through TAS1R2, the sugar-sensing G protein-coupled receptor (GPCR) initially identified in taste perception. Muscle TAS1R2 receptor stimulation by glucose and other agonists induces ERK1/2-dependent phosphorylation and activation of poly(ADP-ribose) polymerase1 (PARP1), a major NAD consumer in skeletal muscle. Consequently, muscle-specific deletion of TAS1R2 (mKO) in male mice suppresses PARP1 activity, elevating NAD levels and enhancing mitochondrial capacity and running endurance. Plasma glucose levels negatively correlate with muscle NAD, and TAS1R2 receptor deficiency enhances NAD responses across the glycemic range, implicating TAS1R2 as a peripheral energy surveyor. These findings underscore the role of GPCR signaling in NAD regulation and propose TAS1R2 as a potential therapeutic target for maintaining muscle health.

Skeletal muscle adapts to various environmental stimuli by coordinating mechanical, hormonal, neuronal, and metabolic pathways. These adaptations are essential for maintaining energy balance and regulating the delicate equilibrium between anabolic and catabolic processes. However, the decline of these mechanisms, often observed in aging and obesity, can result in muscle dysfunction. Central to this scenario is nicotinamide adenine dinucleotide (NAD), an endogenous metabolite that partakes in redox reactions and serves as a substrate for poly(ADP-ribose) polymerases (PARPs) and NAD-dependent deacetylases (sirtuins, SIRT)[1]. Depletion of cellular NAD is closely linked to skeletal muscle atrophy and dysfunction[2]. Therefore, deciphering and

intervening in the mechanisms that control NAD synthesis[3-7] or consumption[8-10] could restore or enhance muscle health. Notably, some of these mechanisms are influenced by signals originating from peripheral nutrient availability[5,6], indicating the existence of nutrient-specific sensors.

G protein-coupled receptors (GPCRs) play a pivotal role in orchestrating cellular functions by integrating signals from both extracellular and intracellular sources. Traditionally activated by hormones and neurotransmitters, an increasing number of specialized GPCRs have emerged that respond to nutrients or endogenous metabolites[11]. These unique GPCRs are distributed in tissues such as

[1]Biological Chemistry & Pharmacology, College of Medicine, The Ohio State University; Columbus, Columbus 43210, USA. [2]Physiology and Cell Biology, College of Medicine, The Ohio State University; Columbus, Columbus 43210, USA. [3]Translational Research Institute, Advent Health, Orlando 32804, USA. ✉e-mail: Georgios.Kyriazis@osumc.edu

the tongue, intestine, and pancreas, where they directly sense energy substrates such as fatty acids, amino acids, or sugars. For instance, sugars like glucose activate sweet taste receptors (STR; TAS1R2/TAS1R3), leading to the release of ATP[12], incretins[13,14], or insulin[15,16] in the corresponding tissues. Consequently, glucose, typically utilized for energy or the generation of regulatory metabolites once inside cells, can independently activate signaling cascades through cell-surface GPCRs. These observations raise the possibility of a broader nutrient-sensing role for this GPCR in other tissues relevant to metabolism. The TAS1R2 subunit of STR provides specificity for sugar sensing[17,18], so to investigate this possibility, we used a Tas1r2-Cre promoter-driven Tdtomato reporter mouse (Td$^{Tas1r2+}$)[19].

Using pharmacological and genetic approaches in vivo and in cell cultures in vitro, we show that *Tas1r2/Tas1r3* are expressed in myofibers and sense ambient glucose to modulate NAD homeostasis. These effects are mediated by the activation of ERK1/2 and the subsequent phosphorylation and activation of PARP1, a main consumer of NAD. Consequently, TAS1R2 signaling inhibition in skeletal muscle increases NAD levels through the suppression of PARP1 activity, causing positive adaptations in muscle fitness.

## Results

### The Tas1r2 and Tas1r3 sweet taste receptors are expressed in skeletal muscle fibers

Assessment of tissues from the Td$^{Tas1r2+}$ reporter mouse showed active expression of *Tas1r2* in skeletal muscles (Fig. 1a and Fig. S1a, b). *Tas1r2* and *Tas1r3* (heterodimeric subunits of STR) were found to be expressed in mouse skeletal muscles (Fig. 1b and Fig. S1c). Available STR

antibodies lacked specificity, so to ensure that their expression was specific to myocytes, we used myogenin-Cre (Myo$^{Cre}$) mice crossed with HA-RiboTag$^{fl/fl}$ mice (HA-RiboTag$^{Myo}$)[20] to selectively immunoprecipitate mRNA from myocytes (skeletal myofibers) (Fig. S1d). We found that STR are specifically expressed in myofibers in vivo (Fig. 1c). Subsequent transcriptomics analysis, using myocyte-specific mRNA from the HA-RiboTag$^{Myo}$ muscles, indicated that *Tas1r2* and *Tas1r3* were among the 139 non-odorant GPCRs expressed in soleus myocytes (Fig. 1d). The mRNA levels of *Tas1r2* and *Tas1r3* were consistent with other expressed GPCRs in myocytes (Fig. 1e and Fig. S2). Consequently, we set to investigate possible contributions of STR-signaling in skeletal muscle function.

### TAS1R2-mediated glucose sensing regulates NAD levels coupled to PARP activity

Due to the involvement of TAS1R2 in glucose-sensing[14,15], we conducted LC/MS-based targeted metabolomics in skeletal muscles of mice with genetic deletion of *Tas1r2* (bKO) or controls (bWT)[18]. We found increased NAD levels in *Tas1r2* ablated muscles, but the NAD precursors[21], NAM and NMN, were similar (Fig. S3a,b). To interrogate muscle-autonomous regulatory mechanisms linked to TAS1R2 signaling, we generated mice with muscle-specific deletion of *Tas1r2* (mKO) and wild type *Tas1r2$^{fl/fl}$* littermate controls (mWT). In agreement with the LC/MS data, we found increased NAD levels in acidic extracts of mKO muscles compared to mWT controls (Fig. 2a) and in differentiated myocyte cultures derived from the same mice (Fig. S3c). Interestingly, the protein levels of phosphoribosyl transferase (NAMPT) were similar in mWT and mKO muscles (Fig. 2b). NAMPT is

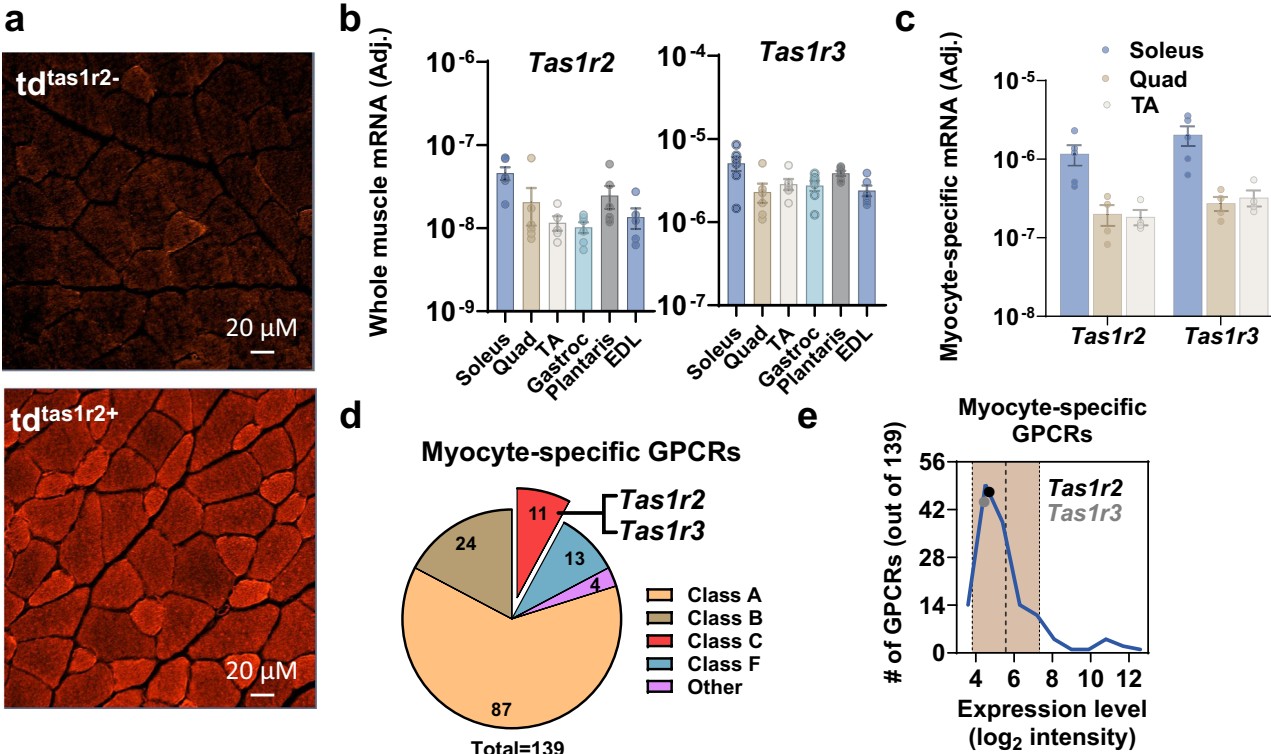

**Fig. 1 | The *Tas1r2* and *Tas1r3* sweet taste GPCRs are expressed in skeletal muscle fibers. a** Immunofluorescence of TdTomato in quadriceps of Tas1r2-Cre:TdTomato-fl/fl mice (Td$^{Tas1r2+}$) and negative control (Td$^{Tas1r2-}$). **b** Expression of the sweet taste receptor genes, *Tas1r2* and *Tas1r3*, in skeletal muscle homogenates (*Tas1r2* Soleus, Quadriceps (Quad), Gastrocnemius (Gastroc) and Plantaris, *n* = 6; Tibialis Anterior (TA) and Extensor Digitorum Longus (EDL) *n* = 5; *Tas1r3 n* = 6). Data are presented as mean ± SEM. **c** Specific expression of *Tas1r2* and *Tas1r3* in myocytes from soleus, Quad, and TA using HA-mRNA pulldown from HA-RiboTag-

Myo mice (Quad and TA *n* = 4; Soleus *n* = 5). Data are presented as mean ± SEM. **d** Distribution of GPCRs (excluding odorants) specifically expressed in myocytes using transcriptomics analysis of mRNA pulldown from HA-RiboTag-Myo mice. *Tas1r2* and *Tas1r3* are among the expressed class-C GPCRs. **e** Frequency plot showing the number of GRCRs at different expression levels using 1 log₂ binning. Shaded area (brown) shows average (middle line) ±1 SD (side lines) expression intensity. *Tas1r2* (black dot) and *Tas1r3* (gray dot) are shown as a reference. Source data are provided as a Source Data file.

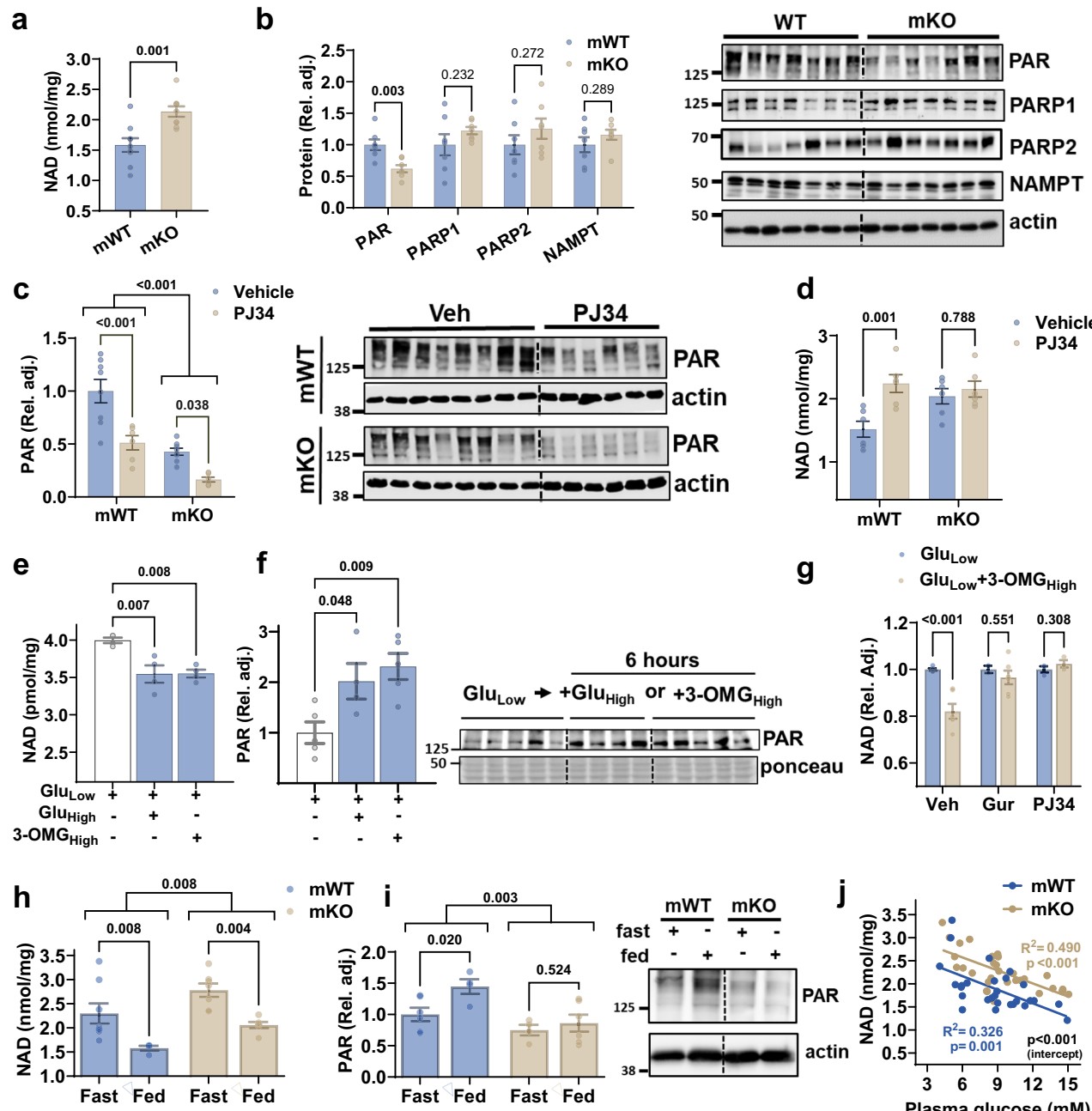

**Fig. 2 | *TAS1R2*-mediated glucose sensing regulates NAD levels coupled to PARP activity. a** NAD concentration in acidic extracts from mWT and mKO muscles ($n = 9$ mice per genotype). Data are presented as mean ± SEM. Two-sided $t$ test. **b** Quantitation and immunoblotting of poly(ADP)-ribosylation (PAR; nuclear), PARP1 (nuclear), PARP2, and NAMPT in mWT and mKO muscles ($n = 7$ mice per genotype). Actin is loading control. Data are presented as mean ± SEM. Two-sided $t$ test. **c** Immunoblotting and quantitation of PAR in nuclear extracts from mWT and mKO muscles in response to 5 days of treatment with the PARP1/2 inhibitor, PJ34 (vehicle-treated, $n = 8$; PJ34-treated, $n = 6$). Data are presented as mean ± SEM. Two-way ANOVA, genotype effect (brackets; $p < 0.001$), Sidak post-hoc effect. **d** NAD concentration in acidic extracts from mWT and mKO muscles in response to 5 days of treatment with the PARP1/2 inhibitor, PJ34 ($n = 6$ mice per genotype and treatment). Data are presented as mean ± SEM. Two-way ANOVA, Sidak post-hoc effect. **e** NAD concentration and (**f**) Immunoblotting and quantitation of PAR in C2C12 cells that were maintained in low glucose (5 mM; Glu Low) and were subsequently spiked with additional glucose (20 mM; Glu High) or 3-OMG (20 mM; 3-OMG High) for 6 h (NAD, $n = 4$; PAR LG and LG + 3OMG, $n = 5$; PAR LG + HG, $n = 4$; sample size are independent experiments). Data are presented as mean ± SEM. One-

way ANOVA, Sidak post-hoc effect. **g** NAD concentration in C2C12 cells that were maintained in low glucose (5 mM) and then were subsequently spiked with additional 3-OMG (20 mM) with or without gurmarin (Tas1r2 inhibitor) or PJ34 (PARP1/2 inhibitor) (Glu 5 mM, $n = 5$; Gur, $n = 2$; PJ34 and 3OMG + PJ34, $n = 3$; 3OMG and 3OMG+Gur, $n = 6$; sample size are independent experiments). Data are presented as mean ± SEM. Two-sided $t$ test. **h** NAD concentration and (**i**) Immunoblotting and quantitation of PAR levels in muscles from mWT and mKO mice subjected to 24 h fast (Fast) followed by 2 h feeding (Fed) (NAD mWT fasted, $n = 8$; NAD mKO fasted, $n = 6$; NAD mWT fed, $n = 4$; NAD mKO fed, $n = 6$; PAR mWT fasted, $n = 6$; PAR mKO fasted, $n = 4$; PAR mWT fed, $n = 4$; PAR mKO fed, $n = 6$). Data are presented as mean ± SEM. One-way ANOVA, Sidak post-hoc effect. $p$-value over grouped brackets indicates genotype effect. **j** Simple linear regression between plasma glucose and muscle NAD concentrations in mWT and mKO mice (mWT, $n = 28$, mKO, $n = 31$). F-tests. Slope $p$-value for each genotype (in color; mWT $p = 0.001$; mKO $p < 0.001$). Elevations contrast $p$-value between mWT and mKO mice (intercept $p < 0.001$). Source data are provided as a Source Data file. Blots were independently repeated in an additional cohort of mice with similar results.

the rate-limiting enzyme in the salvage pathway of NAD biosynthesis[22], recycling NAM, liberated from NAD, to generate NMN[21] (Fig. S3b). Taken together, these findings indicate that TAS1R2 signaling minimally influences the rate of NAD synthesis, suggesting that the increased NAD levels in TAS1R2-deficient muscles may result from a reduction in NAD consumption.

In skeletal muscle, PARP1 is the primary consumer of NAD, utilizing it for protein poly(ADP)-ribosylation (PAR)[1]. Muscles lacking TAS1R2 displayed diminished PAR, with no observable changes in the protein levels of PARP1 or PARP2 (Fig. 2b). The nuclear PAR levels between 115–180 kDa were found to be specifically linked to PARP1 activity, as determined by co-immunoprecipitation (Fig. S3d). We hypothesized that if TAS1R2 influences NAD levels through mechanisms independent of PARP1, the pharmacological inhibition of PARP1 with PJ34, a known enhancer of muscle NAD levels in mice[9,10], would have an additive effect, further elevating NAD levels in mKO mice. PJ34 proportionally suppressed muscle PAR in both genotypes (Fig. 2c). However, while PARP1 inhibition caused an anticipated increase in muscle NAD in mWT mice[10], it failed to further elevate NAD in mKO muscles (Fig. 2d). Thus, reduced PARP1 activity in mKO muscles was sufficient to maximize NAD levels, suggesting that TAS1R2-mediated NAD regulation is coupled, at least in part, to PARP1 activity.

NAD levels in skeletal muscle respond to changes in glucose or energy availability[5,6]. The fact that glucose, the main circulating sugar, is a bona fide TAS1R2 agonist, suggested a physiological link between glycemia and muscle NAD regulation. To test this hypothesis, we acutely stimulated TAS1R2 using glucose or 3-O-methylglucose (3-OMG), a glucose analogue that cannot be metabolized but exhibits the same TAS1R2 affinity as glucose[23,24]. C2C12 cells treated with high glucose or 3-OMG for 6 h exhibited decreased NAD levels (Fig. 2e) and increased PARP1 activity (Fig. 2f) compared to low glucose control. Importantly, the NAD-lowering effect of 3-OMG was prevented by pharmacological inhibition of TAS1R2/TAS1R3 (gurmarin) or PARP1 (PJ34) (Fig. 2g). While differentiated myocyte cultures from mKO muscles exhibited elevated NAD levels compared to mWT cultures (Fig. S3c), stimulation of primary mWT myocytes with glucose did not consistently alter NAD levels. Further investigation is required to elucidate the physiological factors contributing to the NAD insensitivity to glucose manipulations in primary myocytes.

To assess the physiological relevance of the TAS1R2-PARP1-NAD axis in vivo, we subjected mice to fasting and refeeding to induce rapid changes in glycemia (Fig. S3e). Regardless of the feeding state, mKO muscles exhibited higher NAD levels (Fig. 2h) and lower PARP activity (Fig. 2i) than mWT muscles. Refeeding proportionally suppressed muscle NAD levels in both genotypes (Fig. 2h). In contrast, PARP activity increased with feeding in mWT muscles but remained unchanged in TAS1R2-deficient mKO muscles (Fig. 2i), indicating that the modulation of PARP activity in response to glycemia fluctuations relies on functional TAS1R2 signaling. Considering that the TAS1R2 receptor in skeletal muscle is consistently exposed to circulating glucose, we examined and identified a negative correlation between plasma glucose levels and muscle NAD levels in both mWT and mKO mice (Fig. 2j), consistent with the responses observed during fasting and refeeding. Notably, the deficiency of TAS1R2 signaling in mKO muscles amplified NAD responses across the glycemic range, indicating a TAS1R2-dependent regulatory component. Taken together, these findings support a putative link between glucose-induced activation of TAS1R2-PARP1 axis and the regulation of NAD consumption in muscle cells.

## TAS1R2 activates an ERK2-PARP1 signaling axis in skeletal muscle

In neurons, ERK2 can directly phosphorylate and activate PARP1[25–27] independent from DNA damage[28]. Since ERK1/2 is a downstream

mediator of TAS1R2/TAS1R3 signaling[29], we investigated its contribution to the regulation of PARP1 activity in skeletal muscle. The general ERK1/2 inducer, phorbol 12-myristate 13-acetate (PMA), increased PARP activity in C2C12 cells (Fig. S4a), suggesting that, like in neurons, ERK1/2 activation can modulate PARP activity in muscle cells. Acute stimulation of TAS1R2 in vivo through intramuscular injections of sucralose or 3-OMG led to rapid phosphorylation of ERK1/2 in mWT, but not in mKO muscles (Fig. S4b, c), suggesting the presence of a TAS1R2-ERK1/2 signaling axis in skeletal muscle. To further elucidate the specificity of TAS1R2 signaling cascade, we engineered mice with transgenic expression of the human TAS1R2 receptor in the muscles of mKO mice (mTg) and used aspartame, a human-specific TAS1R2 agonist that the mouse TAS1R2 receptor cannot sense[30]. Aspartame injection stimulated ERK1/2 in "humanized" TAS1R2 mTg muscles but not in mWT muscles expressing native mouse TAS1R2 (Fig. 3a). The induction of ERK2 was substantially stronger than ERK1 in both the cytoplasm and the nucleus, where PARP1 is located (Fig. 3a). Indeed, ERK2, but not ERK1, co-immunoprecipitated with PARP1 in muscle nuclear isolates, indicating an interaction (Fig. S4d). In neurons, ERK2 directly phosphorylates PARP1 at the S372/T373 residues[25]. Consequently, ERK2-mediated PARP1 phosphorylation (p-PARP1) and activity (PAR) was increased in muscles from aspartame-dosed mTg mice (Fig. 3b), as detected by an ERK2-specific phospho-PARP1 antibody (Fig. S4d). In contrast, treatment with epidermal growth factor (EGF), which robustly activates ERK1/2 through the EGF receptor tyrosine kinase, did not affect PARP1 phosphorylation or activity in skeletal muscle (Fig. S4e), suggesting specificity of the ERK2-PARP1 axis to TAS1R2 signaling. To determine whether the robust stimulation of TAS1R2 signaling by aspartame can also impact NAD levels in vivo, fasted mWT and mTg mice were subjected to repeated ip. injections of saline or aspartame. Aspartame treatment caused a substantial drop in NAD levels in mTg mice, but aspartame-insensitive mWT mice remained unresponsive (Fig. 3c and Fig. S4f). These findings corroborate the physiological relevance of glucose-induced stimulation of TAS1R2-NAD axis (Fig. 2h–j) and demonstrate that direct stimulation of TAS1R2 in vivo is causatively linked to muscle NAD regulation. Like in muscles in vivo, sucralose activated pERK2 and proportionally increased PAR in C2C12 muscle cells (Fig. 3d, e), but ERK1/2 inhibition with 2-(2-amino-3-methoxyphenyl)-4H-1-benzopyran-4-one (PD98059; PD) (Fig. S4g) blocked sucralose-induced PARP1 phosphorylation (Fig. 3d, f) and activity (Fig. 3d, g). These findings suggest that STR signaling in skeletal muscle modulates PARP1 activity, at least in part, through ERK1/2 activation (Fig. 3h).

## TAS1R2 regulates skeletal muscle mitochondrial capacity and fitness

Increased NAD in skeletal muscle activates the SIRT1-PGC1α axis causing beneficial adaptations in mitochondria[9]. Total acetylation was similar between genotypes (Fig. S5a), but we found reduced PGC1α acetylation (Fig. 4a, b). Taken together with the elevated NAD levels, these findings are indicative of increased SIRT1 activity in mKO muscles. Next, we tested possible effects of the NAD-SIRT1-PGC1α activation on mitochondria content and function. Transmission electron microscopy revealed higher mitochondrial density in myofibers of mKO mice (Fig.4c) and immunoblotting showed elevated levels of electron transport chain (ETC) proteins (Fig. 4d). Consequently, mKO myofibers displayed increased mitochondrial oxidative capacity compared to mWT controls (Fig. 4e). Given the substantial effects on mitochondrial function, we tested the running capacity of mice and found higher endurance in mKO mice (Fig. 4f). These phenotypic adaptations cannot be attributed to differences in body size or growth (Fig. S5b–d), glucose tolerance or insulin sensitivity (Fig. S5e–h), or other secondary effects associated with energy balance and activity (Fig.S5i–l). Collectively, this data indicates that the increased NAD levels resulting from the inhibition of PARP1 activity in TAS1R2-deficient muscles (mKO) contribute to

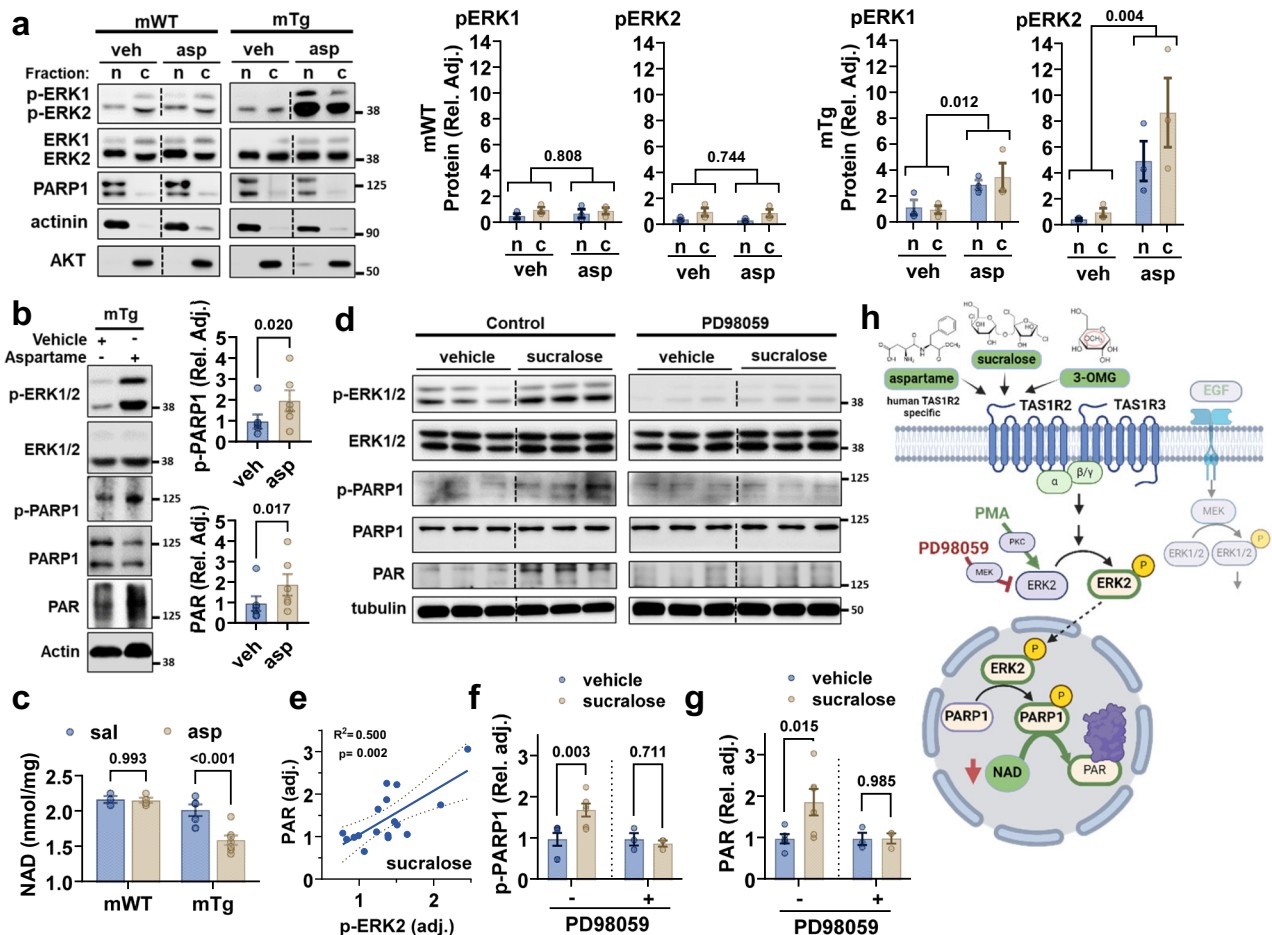

**Fig. 3 | TAS1R2 activates the ERK2-PARP1 axis in skeletal muscle.**
**a** Immunoblotting and quantitation of p-ERK1 and p-ERK2 in response to intra-muscular injection of aspartame (asp) or vehicle (veh) in nuclear (n) or cytoplasmic (c) fractions of mWT and mTg muscles. Relative induction compared to cyto-plasmic vehicle control ($n = 3$ for all groups). Actinin and PARP1 nuclear fraction markers, AKT cytoplasmic fraction marker. Data are presented as mean ± SEM. Two-way ANOVA treatment effect (brackets). **b** Immunoblotting and quantitation of p-PARP1 and PAR in nuclear extracts in response to intramuscular injection of aspartame or vehicle in mTg mice ($n = 6$ in all groups). Data are presented as mean ± SEM. Two-sided $t$ test. **c** NAD concentration in muscles from 24 h fasted mWT and mTg mice treated with i.p. saline (sal) or aspartame (asp) (mWT sal, $n = 3$; mWT asp $n = 4$; mTg sal, $n = 5$, mTg asp, $n = 7$). Data are presented as mean ± SEM. Two-way ANOVA (interaction p = 0.013), Sidak post-hoc effect. **d** Immunoblotting of p-ERK 1/2, ERK 1/2, p-PARP1, PARP1, and PAR in C2C12 cells treated with sucralose or vehicle, with or without the ERK1/2 inhibitor, PD98059. **e** Simple linear

regression of p-ERK2 with PAR in sucralose-treated C2C12 cells ($n = 17$). F-test. Slope $p$-value. **f** Immunoblotting of p-PARP1 and (**g**) Immunoblotting of PAR in C2C12 cells treated with sucralose or vehicle, with or without the ERK1/2 inhibitor, PD98059 (vehicle control, $n = 6$; vehicle sucralose, $n = 6$; PD98059 control, $n = 3$; PD98059 sucralose, $n = 3$). Data are presented as mean ± SEM. Two-sided $t$ test. **h** Schematic of TAS1R2 signaling cascade. Activation of STR (TAS1R2/TAS1R3) with agonists such as 3-OMG, sucralose, or aspartame (human TAS1R2-specific) increa-ses the phosphorylation of ERK2 (and ERK1) in the cytoplasm and nucleus. Acti-vated ERK2 interacts with and directly phosphorylates PARP1 leading to its activation and increased PAR. PMA is an activator and PD98059 an inhibitor of ERK1/2, respectively. Panel (h) was created with BioRender.com released under a Creative Commons Attribution-NonCommercial-NoDerivs 4.0 International license. Source data are provided as a Source Data file. Blots were repeated twice with similar results.

noteworthy functional adaptations in mitochondria and overall muscle fitness (Fig. 4g).

## Discussion

All cells possess glucose-sensing mechanisms that link energy avail-ability to cellular processes governing homeostasis. Typically, these mechanisms rely on glucose uptake and subsequent transformation into metabolites detectable by intracellular sensors[31]. In this study, we elucidate the relationship between glucose sensing and NAD home-ostasis in skeletal muscle, emphasizing the critical role of TAS1R2, the GPCR traditionally associated with sweet taste perception[17]. Our find-ings reveal that TAS1R2 directly regulates NAD levels, underscoring its significance in linking peripheral energy availability to cellular pro-cesses controlling homeostasis.

We demonstrate the expression of *Tas1r2* and *Tas1r3* in myofibers, suggesting that sweet taste receptors have a biological function in

skeletal muscle. Importantly, their expression levels align with other relevant GPCRs, indicating a functional role in sensing and responding to circulating energy substrates. Using various agonists, we show that activation of TAS1R2 signaling negatively regulates NAD levels by increasing PARP1 activity. In skeletal muscle, PARP1 is the primary consumer of NAD, utilizing it for protein poly(ADP)-ribosylation (PAR)[1]. Consequently, pharmacological or genetic inhibition of TAS1R2 attenuates PARP1 activity, causing sustained increases in NAD levels. Mechanisms or strategies that boost NAD levels can induce beneficial adaptations in skeletal muscle[32]. In part, this is because NAD is also a substrate for SIRT1, so lowering PARP1 activity can redirect NAD toward SIRT1-mediated utilization[21]. We found increased SIRT1 activity in mKO muscles associated with beneficial adaptations in mitochon-drial capacity and running endurance. These findings demonstrate that elevated NAD levels in TAS1R2 deficient muscles can translate into significant physiological adaptations. Direct inhibition of PARP1, the

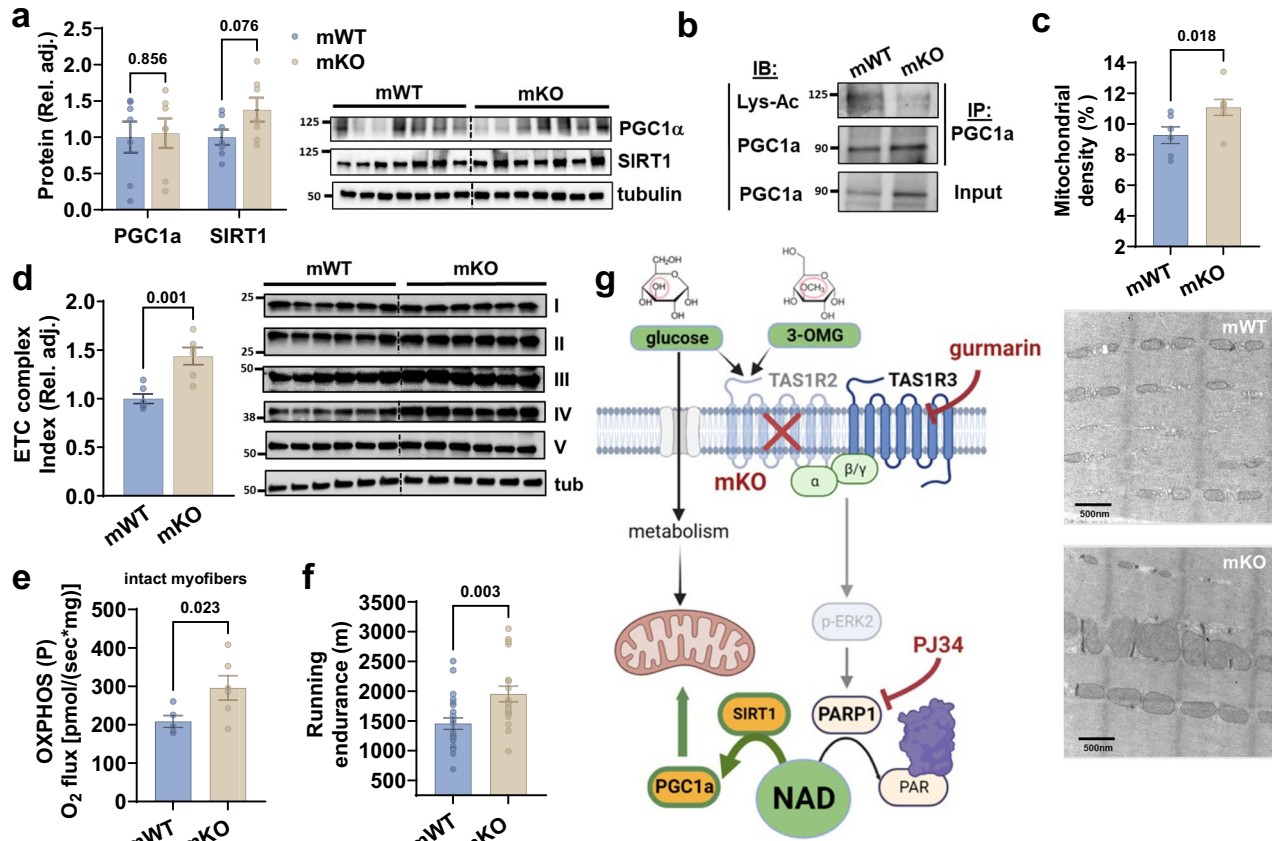

**Fig. 4 | TAS1R2 regulates skeletal muscle mitochondrial capacity. a** Quantitation and immunoblotting of PGC1α and SIRT1 in mWT and mKO muscles (mWT $n = 7$; mKO $n = 7$). Tubulin is loading control. Data are presented as mean ± SEM. Two-sided $t$ test. **b** Immunoblotting (IB) of acetylated PGC1α (Lys-AC) in mWT and mKO muscle lysates immunoprecipitated (IP) with total PGC1α. **c** Mitochondrial density and structure of mWT and mKO soleus muscle using transmission electron microscopy (TEM) (mWT $n = 6$; mKO $n = 7$). Data are presented as mean ± SEM. Two-sided $t$ test. **d** Immunoblotting and quantitation of mitochondrial electron transport chain (ETC) proteins in mWT and mKO muscle (mWT $n = 6$; mKO $n = 6$). Data are presented as mean ± SEM. Two-sided $t$ test. **e** Respiratory capacity in the ADP-activated state of oxidative phosphorylation (OXPHOS P) of intact muscle fibers from mWT and mKO mice (mWT $n = 5$; mKO $n = 6$). Data are presented as mean ± SEM. Two-sided $t$ test. **f** Treadmill running endurance of mWT and mKO mice using constant intensity to exhaustion (mWT $n = 22$; mKO $n = 19$). Data are presented as mean ± SEM. $t$ test. **g** Genetic (mKO) inhibition of TAS1R2 reduces PARP1 activity, increasing the availability of NAD for SIRT1. This causes subsequent beneficial effects on mitochondrial function and muscle fitness. Direct inhibition of PARP1/2 with PJ34 can elicit similar effects. Panel (**g**) was created with BioRender.com released under a Creative Commons Attribution-NonCommercial-NoDerivs 4.0 International license. Source data are provided as a Source Data file. Blots were repeated twice with similar results.

downstream effector of TAS1R2 signaling, also activates the NAD-SIRT1 axis leading to similar beneficial effects, including mitochondrial content and function, and running endurance[9,10]. However, these studies targeting PARP1 inhibition did not demonstrate or propose any physiological context or molecular signal that mediates these responses[9,10]. Therefore, TAS1R2 and PARP1 loss-of-function in mice display similar phenotypic attributes, further corroborating a functional coupling between TAS1R2 and PARP1 activity. The adaptive effects resulting from TAS1R2 deficiency in skeletal muscle are comparable to other approaches that positively affect NAD levels, such as NAD precursor supplementation[2,3,33] or exercise training[4,6,34,35], all directed towards promoting NAD synthesis. However, enhancing NAD synthesis through NAMPT overexpression in skeletal muscle proved insufficient to promote mitochondrial adaptations[36], suggesting a dynamic interplay between complex mechanisms that control NAD bioavailability. In light of this, the adaptations observed in our mouse models imply that regulating NAD consumption might yield greater benefits for skeletal muscle than solely focusing on promoting its synthesis.

Most notably, our study establishes TAS1R2 as a glucose-sensing GPCR with independent influence on NAD levels in skeletal muscle.

Fluctuations in ambient glucose induced by fasting-refeeding cycles or glucose restriction are traditionally associated with changes in NAD biosynthesis, underscoring an intimate link between energy availability and the modulation of NAD homeostasis[5,6]. These effects have been attributed to shifts in energy metabolism and the activation of intracellular energy sensors that depends on nutrient metabolism. However, our findings suggest that TAS1R2-mediated glucose sensing can independently alter NAD levels through extracellular energy sensing. This is particularly relevant given that the physiological range of blood glucose concentrations aligns with the affinity values of glucose for TAS1R2[37]. Therefore, in addition to the modulation of NAD synthesis by metabolic sensors through the transcriptional regulation of NAMPT[5,6], TAS1R2 receptor-mediated glucose sensing concurrently targets NAD consumption through the regulation of PARP1 activity. Targeting NAD synthesis and consumption through two distinct mechanisms, one relying on glucose uptake and metabolism[6] and the other on receptor-mediated peripheral glucose sensing, may be necessary for critical adjustments of the NAD pool accompanying large energy or nutrient shifts. Considering that intracellular energy sensors, such as AMPK[6,35], can be equally influenced by the metabolism of glucose or any other macronutrient, a mechanism involving nutrient-specific GPCR

signaling can be rapidly deployed following changes in the circulating concentrations of the corresponding nutrient. This may confer a physiological advantage for the modulation of NAD that depends on both general energy cues and nutrient-specific ones. Although this notion requires further investigation, skeletal muscle also expresses GPCRs that sense fatty acids[38] and amino acids[39] which may contribute to this or other relevant sensory processes.

Another finding of our studies is the identification of an ERK1/2-PARP1 signaling axis in skeletal muscle. Specifically, ERK1/2 activation downstream of TAS1R2 stimulation directly phosphorylates and activates PARP1, leading to NAD utilization. ERK1/2 have been implicated in the regulation of skeletal muscle maintenance[40] or development[41], but their direct effects on muscle bioenergetics are unclear[42]. The ERK2-PARP1 axis was shown to be selective, as robust activation of ERK1/2 through the EGF tyrosine kinase receptor had no effect on PARP1 phosphorylation of activity. Therefore, the selective interaction between ERK1/2 and PARP1 underscores the specificity of the TAS1R2 signaling cascade in skeletal muscle in response to relevant extracellular stimuli. This concept is aligned with previous reports showing that receptor-mediated amino acid sensing in skeletal muscle regulates mTORC1 through ERK1/2 signaling[39,43]. Taken together, these findings raise the possibility that myocytes integrate GPCR-dependent signals derived for all classes of circulating macronutrients.

In summary, our study uncovers a GPCR-dependent signaling mechanism that fine tunes skeletal muscle NAD bioavailability in response to changes in extracellular glucose. We postulate that TAS1R2 may function as a peripheral energy surveyor, conveying signals to myocytes about shifts in energy availability. Consequently, chronic TAS1R2 deficiency is likely perceived as prolonged energy deprivation eliciting NAD-dependent adaptations akin to those observed under similar conditions[1]. This mechanism offers insights into the intricate interplay between nutrient sensing and cellular adaptations and unveils possible therapeutic approaches. Given that GPCRs are established targets for pharmacological interventions[44], the inhibition of TAS1R2 receptor could benefit individuals with compromised muscle health.

## Methods

### Animals
All animal procedures were performed under the approval of The Ohio State University institutional animal care and use committee (IACUC). Mice were housed on a 12 h light/dark cycle with free access to water and standard diet. Male mice were used for experimental purposes only after reaching sexual maturity at 12-16 weeks of age unless otherwise stated.

*Tas1r2* floxed mice (Tas1r2$^{fl/fl}$) were generated from the KOMP Repository (#CSD25803). *Tas1r2* iCre knock in mice (Tas1r2$^{iCre}$) were previously generated as described[19]. Human *TAS1R2* transgenic mice (hTAS1R2$^{fl-stop}$) were created by InGenious Targeting Laboratory (Ronkonkoma, NY) by placing a cDNA cassette containing a synthetic CAG promoter, a floxed NEO stop cassette, the consensus human TAS1R2 cDNA, an IRES-EGFP, and a BGH polyA on the mouse *ROSA26* locus. Whole body *Tas1r2* knockout mice (bKO) were a gift from Dr. Zuker[18]. *Myogenin*-Cre mice (Myo$^{Cre}$) were a gift from Dr. Olson[45]. The above mouse lines (hTAS1R2$^{fl-stop}$, Tas1r2$^{fl/fl}$, Tas1r2$^{iCre}$, and Myo$^{Cre}$) were backcrossed and then maintained to pure C57BL6/6 J strain (using Jax# 000664). RiboTag-Rpl22$^{fl/fl-HA}$ (RiboTag$^{fl/fl}$) (Jax# 029977) and tdTomato mice (Td$^{fl/fl}$) (Jax#007914) were obtained from the Jackson Lab (Bar Harbor, ME).

Td$^{fl/fl}$ mice were crossed with Tas1r2$^{iCre}$ mice to generate mice (Td$^{Tas1r2}$) that express Tdtomato in Tas1r2 positive cells[19]. RiboTag$^{fl/fl}$ mice were crossed with Myo$^{Cre}$ mice to generate mice (HA-RiboTag$^{Myo}$) that express Rpl22-HA in myogenin-positive cells (myocytes).

Tas1r2$^{fl/fl}$ mice were crossed with Myo$^{Cre}$ mice to generate mice (mKO) with muscle-specific deletion of *Tas1r2*. mKO mice were crossed with hTAS1R2$^{fl-stop}$ mice to generate mice (mTg) with muscle-specific transgenic overexpression of human *TAS1R2* without endogenous *Tas1r2*. We used a specific breeding scheme to obtain wild-type control (mWT), mKO, and mTg congenic littermates: Congenic male mice carrying one Myo$^{Cre}$ allele, one hTAS1R2$^{fl-stop}$ allele and two Tas1r2$^{fl/fl}$ alleles were crossed with congenic female mice carrying two Tas1r2$^{fl/fl}$ alleles. Male offspring carrying two Tas1r2$^{fl/fl}$ alleles were used as wild-type controls (mWT). Male offspring carrying one Myo$^{Cre}$ allele and two (floxed out) Tas1r2$^{fl/fl}$ alleles were used for muscle-specific genetic deletion of *Tas1r2* gene (mKO). Male offspring carrying one Myo$^{Cre}$ allele, two (floxed out) Tas1r2$^{fl/fl}$ alleles, and one (floxed STOP) hTAS1R2$^{fl-stop}$ allele were used as muscle-specific transgenic expression of *hTAS1R2* gene without the endogenous mouse *Tas1r2* gene (mTg).

### Mouse genotyping
Mice were ear-punched at weaning and punches were digested with 100 μL of 50 mM NaOH for 1 h at 95 °C. Samples were then cooled at room temperature, spun, and neutralized with 20 μL of 1 M Tris pH8 and 100μL of water. Clear homogenates were subsequently genotyped by PCR and electrophoresis using specific primers. Primer sequences and genotyping details can be found in the Supplementary Information.

### Mouse in vivo treatments
*PJ34 treatment:* Administration of 10 mg/kg PJ34 (Cayman Chemicals, Ann Arbor, MI) was performed IP twice a day for 5 days before the tissue harvest. *Sweetener injections:* For intramuscular sweetener injections, 5 h fasted mice were anesthetized with 165 mg/kg pentobarbital (Sagent, Schaumburg, IL) and 10μL of sucralose (100 mM) (Sigma-Aldrich, Saint Louis, MO), 3-o-methylglucose (30 mM) (Sigma-Aldrich, Saint Louis, MO), or aspartame (5 mM) (Cayman Chemicals, Ann Arbor, MI) were injected with a 20μL Hamilton syringe (Hamilton Company, Reno, NV) in the vastus medial using the contralateral leg as saline-injected control. Muscles were dissected exactly 10 min after each injection and snap-frozen in liquid nitrogen. *Fast/refed treatment:* Mice were divided into cohorts of 4 mice and randomized between fasted and refed. Each day, a cohort was fasted starting 7am for a total of 24 h. Mice were kept fasted or were refed in a staggered fashion, so each harvest was performed exactly 2 h after the additional fasting or refeeding. *Aspartame treatment:* Mice were divided into cohorts of 4 mice and randomized between Saline and Aspartame. Each cohort was fasted for 20 h and mice were then injected IP with 200 μL saline (Pfizer, New York, NY) or 200μL of 7.5 mg/ml aspartame (60 mg/kg) (Cayman Chemicals, Ann Arbor, MI). Mice were injected six times (360 mg/mg aspartame) spanning a 4 h period and harvested right after.

### Harvest and tissue processing
All harvests were performed between 1 pm and 2 pm after a 5 h fast, unless otherwise stated. All tissues were collected in vivo under 165 mg/kg pentobarbital anesthesia (Sagent, Schaumburg, IL) using heat pads to sustain body temperature. Muscles were harvested tendon to tendon starting with the left leg and immediately snap-frozen in liquid nitrogen. Samples for RNA extraction were stored at -80 °C until use. Samples for protein determination and metabolomics were lyophilized overnight at -50 °C and 0.5mbar and then pulverized in 2 ml Sarstedt tubes (Newton, NC) containing 2 RNase-free plastic beads (OMNI, Kennesaw GA) in a Precellys homogenizer (Bertin Technologies, Rockville, MD) at 0 °C × 7200RPM × 20 s × 15 s pause × 3 cycles.

### Primary myocyte isolation and culture
Mouse hindlimb muscles were dissected in PBS (Gibco-Life Technologies, Grand Island, NY), sliced into 1 mm pieces in HBSS (Gibco-Life Technologies, Grand Island, NY), and digested for 45 min at 37 °C using 400U/ml collagenase (Gibco-Life Technologies, Grand Island, NY) and

0.08 U/ml dispase II (Roche, Germany). Tissues were then triturated by aspiration with a 1 ml pipette and incubated for additional 45 min before the reaction was stopped by addition of pre-warmed pre-plating medium (PPM), consisting of 90% low glucose DMEM (Sigma-Aldrich, Saint Louis, MO), 10% FBS, 1x glutamax, 1% penn-strep (all Gibco-Life Technologies, Grand Island, NY). Cells were strained, spun at 300 g, re-suspended in PPM, and incubated into a 100 mm tissue dish for 3 h. Myoblasts were then released from the culture dish by gentle agitation leaving the adherent fibroblasts behind and the medium was spun at 300 g. Pelleted cells were re-suspended in growth medium (GM), consisting of 80% Ham's F-10, 20% FBS, 1x Glutamax (Gibco-Life Technologies, Grand Island, NY), 2.5 ng/ml bFGF, 10 ng/µL EGF, 1ug/ml insulin, 0.39 mg/ml dexamethasone (all Sigma-Aldrich, Saint Louis, MO), 1% penn-strep (Gibco-Life Technologies, Grand Island, NY). Cells were incubated on collagen-coated T25 flasks, and expanded for 5-6 days until clusters of 2-5 cells appeared. Cells were then purified for a second time (trypsinized in 1:1 PBS, pre-plated with PPM, separated from fibroblasts by agitation, spun at 300 g, resuspended in GM) and expanded from T25 to T75 flasks. Cells were differentiated at passage 6 by seeding 50000 cells into collagen-coated 24-well plates and switching to differentiation media (DM), consisting of 98% high glucose DMEM (Sigma-Aldrich, Saint Louis, MO), 2% horse serum, 1x Glutamax, 1% pen-strep (all Gibco-Life Technologies, Grand Island, NY) 24 h after. NAD was determined after 6 days of differentiation.

## C2C12 culture and studies
C2C12 (ATCC CRL-1772) were cultured in high glucose DMEM (Sigma-Aldrich, Saint Louis, MO) with 10% fetal bovine serum and 1% Penn/Strep (all Gibco-Life Technologies, Grand Island, NY) using 75cm² flasks. Cells were passaged in 24-well plates for NAD or 6-well plates for signaling studies at a density of 52632 cells/cm² and media was changed to high glucose DMEM (Sigma-Aldrich, Saint Louis, MO) with 2% horse serum and 1% Penn/Strep (all Gibco-Life Technologies, Grand Island, NY) upon confluence to start differentiation. For the assessment of NAD cells were switched to 5 mM glucose DMEM (Sigma-Aldrich, Saint Louis, MO) 2% horse serum and 1% Penn/Strep at day 4 of differentiation (all Gibco-Life Technologies, Grand Island, NY). After 2 days, the acute effects of adding glucose (20 mM) (Gibco-Life Technologies, Grand Island, NY), 3-o-methylglucose (20 mM) (Sigma-Aldrich, Saint Louis, MO), PJ-34 (1uM) (Cayman Chemicals, Ann Arbor, MI) or gurmarin (30ug/ml) (MyBioSource, San Diego, CA) were evaluated by spiking stock volumes in the wells for 6 h before harvesting. For signaling experiments, C2C12 cells at day 6 of differentiation were equilibrated in EBSS (Gibco-Life Technologies, Grand Island, NY) for 10 min and subsequently incubated with 2x stocks in EBSS to achieve 10 mM sucralose (Sigma-Aldrich, Saint Louis, MO), 10µM PD98059 (Cayman Chemicals, Ann Arbor, MI), 10 ng/ml EGF (Sigma-Aldrich, Saint Louis, MO), or 0.1 mM PMA (Sigma-Aldrich, Saint Louis, MO).

## Oxygen consumption in isolated fibers and primary cultures
Measurement of mitochondrial function (oxygen consumption) in muscle fibers and cells was performed as described[46]. Oxygen consumption in saponin-permeabilized fibers was performed using Oxygraph 2 K (Oroboros, Austria). Maximal respiration supported by electron flux was measured with the addition of ADP (500 mM) after stabilization on saturating pyruvate-malate (2 M). Steady-state O2 flux for was determined and normalized to fiber dry weight using Datlab 6 software (Oroboros).

## Mouse body metrics (podikometry)
Body composition was measured in duplicate using an EchoMRI instrument, calibrated daily with 40.5 g of canola oil, and percent composition was calculated after subtraction of free water. Body length and tibia length were measured at harvest using pre-calibrated

Traceable digital calipers (Fisher Scientific, Waltham, MA). Liver weight was measured immediately during the dissection with a 0.1 mg precision.

## Glucose and insulin homeostasis
Glucose tolerance tests were performed intraperitoneally (i.p.GTT) or intragastrically (ig.GTT) with 1 g/kg glucose in 5 h fasted mice. Insulin tolerance test (ITT) was assessed in 5 h fasted mice with 0.5U/kg insulin (Humulin R U-100; Eli Lilly, Indianapolis, IN). Plasma glucose was measured from the tail vein using AlphaTRAK 2 glucometers (Zoetis, Parsippany, NJ). Plasma glucose and insulin was measured in ad libitum mice (9am). For insulin measurement, blood was collected and centrifuged immediately at 2000g in EDTA-treated tubes and analyzed via ELISA (ChrystalChem ultrasensitive insulin kit; Elk Grove Village, IL). The quantitative insulin sensitivity check index (QUICKI) was calculated as the inverse of the sum of the logarithms of insulin (µU/ml) and glucose (mg/dL).

## Energy balance and activity
An automated phenotyping homecage system (TSE PhenoMaster; TSE Systems, Germany) was used to perform behavioral and metabolic tests in single-caged mice. Phenotypic measurements of activity, food, and water intake and expired $O_2$ and $CO_2$ were performed ad libitum in cohorts of male mWT and mKO mice. RER and expenditure were automatically calculated from the software (TSE PhenoMaster 15.7.6.2) using the values of expired gases. Data was analyzed by regression following standardized guidelines using jamovi 2.2.5[47].

## Running endurance (treadmill)
Running endurance was performed between 12 and 2 p.m. in 5 h fasted male mice using an Exer 6 treadmill (Columbus Instruments, Columbus OH). All mice were accustomed to the treadmill for at least 2 sessions of acclimatization. Endurance was tested with a constant intensity protocol, consisting of 5 min–0°–10 m/min, 5 min–0°–20 m/min, and then 10°–20 m/min until exhaustion.

## RNA isolation, gene expression and analysis
Frozen tissue samples were homogenized with 1 mL of TRIzol (Invitrogen-Thermo Fisher Scientific, Waltham, MA), 3 2.8 mm plastic beads (OMNI, Kennesaw GA), and 1 0.05 mL scoop of 0.05 mm glass beads (Next Advance, Troy, NY) at 0 °C × 6800RPM × 20 s × 10s pause × 9 cycles. Homogenization was repeated 5 times. Samples were separated from beads and mixed with 200 µL of chloroform (Sigma-Aldrich, Saint Louis, MO) in 1.5 mL tubes. Samples were centrifuged at 12,000$g$ × 4 °C × 15 min. Clear phase was separated, and ethanol (200 proof; Fisher Scientific, Waltham, MA) was added 1:1 with clear phase. A Zymo Research Direct-zol RNA Microprep kit (Irvine, CA) was used for purification. RNA concentration was measured using NanoDrop (Thermo Fisher Scientific, Waltham, MA). 1µg RNA was reverse transcribed using the High-Capacity cDNA Reverse Transcription kit (Applied Biosystems-Thermo Fisher Scientific, Waltham, MA). cDNA was brought to 100µL and 2µL of sample per well was used in real time PCR using SYBR chemistry (Bio-Rad, Hercules, CA). The cycle thresholds were compared to the *Rn18s* gene and gene expression was expressed in arbitrary units as $2^{-\Delta Ct}$. Primer sequences can be found in the Supplementary Information.

## Ribotag RNA isolation and analysis
Muscle samples of RiboTag[Myo] mice were harvested, washed in PBS with 100 µg/ml cycloheximide (Millipore-Sigma, Burlington, MA), and frozen in liquid nitrogen. Frozen samples were pulverized and homogenized in 1 mL of cold polysome buffer (20 mM Tris pH 7.4, 10 mM MgCl, 200 mM KCl; all Invitrogen-Thermo Fisher Scientific, Waltham, MA; 2 mM DTT, 1% Triton X-100, 100 µg/ml cycloheximide; all Millipore-Sigma, Burlington, MA) using 2 mL Dounce homogenizer

(Wilmad, Vineland, NJ). Homogenates were transferred to a 1.5 ml RNase-free tube and centrifuged at 17,000 × g, 4 °C, for 10 min. The supernatant was brought to a fresh 1.5 mL tube and 100 µL of sample was saved as input. The remaining supernatant was incubated with 4 µL of anti-hemagglutinin antibody for 3–6 h at 4 °C with constant rotation. 50 µL of AG Magnetic Beads (Pierce-Thermo Fisher Scientific, Waltham, MA) were washed 5 times with polysome buffer, re-suspended into 50 µL of polysome buffer, and added to the samples. Following overnight rotation, the samples were placed on a magnetic rack and washed 5x with 500 µL of high salt buffer (20 mM Tris pH 7.4, 10 mM MgCl, 300 mM KCl; all Invitrogen-Thermo Fisher Scientific, Waltham, MA; 2 mM DTT, 1% Triton X-100; all Millipore-Sigma, Burlington, MA) before elution with 300 µL of TRIzol (Invitrogen-Thermo Fisher Scientific, Waltham, MA). The supernatant was mixed with an equal volume of ethanol and RNA was immediately extracted using a purification kit (Zymo Research, Irvine, CA). Reverse transcription and real time PCR were performed thereafter with our standard protocol. Gene enrichment was calculated as the ratio between elution and input expression. Associations with cell markers were evaluated through principal component analysis of eluate gene expressions after normalization with *Rn18s*.

## Transcriptomics
RNA eluates from RiboTag$^{Myo}$ muscle tissue pulldown were send to the OSU Genomics Core for transcriptomic profiling using an Affymetrix mouse Clariom S hybridization array (Thermo Fisher Scientific, Waltham, MA). Samples were processed per the manufacturer's protocol for a Clariom S Pico assay after a quality and quantity control with an Agilent TapeStation 4200 (Agilent, Santa Clara, CA) and a NanoDrop (Thermo Fisher Scientific, Waltham, MA). The resulting hybridization intensities were processed with Transcriptome Analysis Console 4.0 using the default configuration over the Clariom S Mouse updated transcript version 2.

## Immunohistochemistry and tissue fluorescence
For TdTomato tissue fluorescence, muscles were cryo-protected in 30% sucrose overnight, then frozen in OCT (Thermo Fisher Scientific, Waltham, MA), sectioned, mounted (Prolong Gold Antifade Mountant; Thermo Fisher Scientific, Waltham, MA), and imaged using Zeiss LSM 900 confocal (Zeiss, Germany) with a 581 nm filter and Zen 3.0 Zeiss Confocal software.

## Electron Microscopy
Transmission electron microscopy imaging of muscle samples was performed by the OSU imaging core. Soleus samples were specially chosen for their fast fixation, thus preserving mitochondrial ultrastructure. Samples were dissected and fixed in 2.5% glutaraldehyde (Sigma-Aldrich, Saint Louis, MO) in 0.1 M phosphate buffer. Samples were postfixed with 1% osmium tetroxide and then *en bloc* stained with 1% aqueous uranyl acetate, dehydrated in a graded series of ethanol, and embedded in Eponate 12 epoxy resin (Ted Pella Inc., Redding, CA). Ultrathin sections were cut with a Leica EM UC7 ultramicrotome (Leica microsystems Inc., Deerfield, IL) and collected on copper grids. Images were acquired with an FEI Technai G2 Spirit transmission electron microscope (Thermo Fisher Scientific, Waltham, MA) operating at 80 kV, and a Macrofire (Optronics, Inc., Chelmsford, MA) digital camera and AMT image capture software. For random sampling during imaging, at least 3 myofibers per subject were identified and then each was imaged in at least 4 random locations. The process was repeated at 11500x and 34000x.

## NAD cyclic assay
NAD was determined by a cycling assay based on alcohol dehydrogenase after the acidic destruction of NADH[48]. 500 µL of ice-cold 0.6 M perchloric acid (Sigma-Aldrich, Saint Louis, MO) was added to pelleted cells or to 2-3 mg of lyophilized tissue placed in 500 µL skirted tubes with 5 mg of glass beads (Next Advance, Troy, NY). Samples were homogenized at 0 °C in a Precellys homogenizer (Bertin Technologies, Rockville, MD) with 3 cycles of shaking at 7200RPM × 20 s and 15 s pause, repeated 4 times. The homogenates were centrifuged at 12000 g during 20 min and 4 °C, and 300 µL of acidic extract was saved in a new ice-cold 1.5 ml tube and the acidic pellet was saved for protein determination. Samples were then diluted 1:10 (cells) or 1:100 (tissues) with 100 mM PBS (Gibco-Life Technologies, Grand Island, NY) right before starting the assay. The assay was performed in triplicate in black opaque 96 well plates containing 5 µL of sample and standards and 95 µL of freshly-made cycling assay. The cycling assay consisted on 100 mM PBS (Gibco-Life Technologies, Grand Island, NY), 0.1% BSA (Sigma-Aldrich, Saint Louis, MO), 10 mM NAM (Sigma-Aldrich, Saint Louis, MO), 2% ethanol (Fisher Scientific, Waltham, MA), 10 µM FMN, 20 µM resazurin, 100 µg/µL ADH, 10ug/ml diaphorase (all from Sigma-Aldrich, Saint Louis, MO). Fluorescence was recorded at 530/590 for 30 min. NAD from muscle samples was normalized by mg of dry tissue. For cell determinations, the total protein per well was used to normalize NAD. The acidic cell pellet was centrifuged again at 12000 g during 20 min and 4 °C and any remaining supernatant was discarded. The pellet was then neutralized with 50 µL 1 M NaOH and homogenated at 7200RPM x 20 s three times or until solubilization. The basic homogenate was then spun and neutralized with 250 µL 100 mM PBS before measuring protein with a BCA kit (Thermo Fisher Scientific, Waltham, MA).

## Muscle metabolomics
Targeted, quantitative metabolomics for nucleotides were performed from lyophilized gastrocnemius muscle as previously described[49–51].

## Protein extraction
4–6 mg of lyophilized powdered tissue was weighed in 2 ml Sarstedt tubes and protein was extracted with 200 µL of RIPA buffer (Thermo Fisher Scientific, Waltham, MA) supplemented with protease and phosphatase inhibitors (Roche, Germany). Samples were placed 10 min on ice and then homogenized with 3 cycles of shaking at 7200RPM × 20 s and 15 s pause in a Precellys (Bertin Technologies, Rockville, MD). After spinning at 12000 g during 30 min, supernatants were placed in new tubes and protein was measured immediately with a BCA kit (Thermo Fisher Scientific, Waltham, MA). Samples were then brought to 2 mg/ml with supplemented RIPA buffer and 6x reducing buffer (Alfa Aesar- Thermo Fisher Scientific, Waltham, MA) and stored at −20 °C if not used immediately.

## Immunoblotting
Samples were heated for 10 min at either 65 °C (samples probed for parylation or mitochondrial proteins) or 85 °C and 10–30 µg of protein per lane was loaded into standard 7-10% reducing acrylamide gels for electrophoresis. Protein transfer onto PVDF (for mitochondrial proteins) or nitrocellulose membrane was performed in Towbin buffer at 100 V for 90 min inside a cold 4 °C room. After the transfer, blots were Ponceau-stained, cut into pieces to match the molecular weight of a target and a housekeeping protein, de-stained, and blocked with 5% dehydrated milk (Bio-Rad, Hercules, CA) in TBST buffer. Blots were then probed overnight at 4 °C with corresponding primary antibodies and continuous rocking. After 3 washes in TBST, blots were incubated for 1 h with secondary antibodies, washed again and visualized with SuperSignal reagent (Thermo Fisher Scientific, Waltham, MA) in a Bio-Rad ChemiDoc imaging system. Antibody details and blot scans can be found in Supplementary Information.

## Nuclear isolation
Lyophilized, pulverized muscle powder (12–15 mg) was resuspended in 300 µL PBS containing protease inhibitors (Sigma-Aldrich, Saint Louis,

MO) and phosphatase inhibitors (Roche, Germany), and was homogenized via dounce homogenizer. Homogenate was centrifuged at 1900 × *g*, 4 °C for 15 min, and supernatant was isolated as cytosolic fraction. The pellet was further treated with proprietary solutions for nuclear isolation (NE-PER, Thermo Fisher Scientific, Waltham, MA), as follows. CER I buffer and NER buffer were supplemented with protease inhibitors (Sigma-Aldrich, Saint Louis, MO) and phosphatase inhibitors (Roche, Germany). The pellet was resuspended in 150 µL CER I buffer and incubated 10 min on ice. Buffer CER II (8.25 µL) was added to each sample, incubated 1 min on ice, then centrifuged at 16,000 x g, 4 °C for 5 min. Supernatant was collected and added to the cytosolic fraction. Pellet was resuspended in 75 µL NER and incubated on ice for 40 min, with vortexing every 10 min. Samples were centrifuged at 16,000 x g, 4 °C for 10 min. The supernatant was collected as a nuclear fraction. Sample protein concentration was determined via BCA assay (Thermo Fisher Scientific, Waltham, MA).

### Immunoprecipitation
Protein samples (450 µg of each) were diluted to 1 µg/µL in RIPA buffer (Thermo Fisher Scientific, Waltham, MA) with protease inhibitors (Sigma-Aldrich, Saint Louis, MO) and phosphatase inhibitors (Roche, Germany). To pre-clear samples, 50 µL Protein A/G-conjugated magnetic beads (Pierce-Thermo Fisher Scientific, Waltham, MA) were added to each sample and incubated for 60 min at 4 C while rotating. Beads and supernatant were separated magnetically, and 50 µL of supernatant was isolated as an input sample. The remaining supernatant was incubated with 4 µL PARP antibody overnight at 4 °C with rotation. Protein A/G-conjugated beads (50 µL) were added to each sample and incubated at 4 °C for 2 hours. Beads and supernatant were separated magnetically, and supernatant was discarded. Beads were washed 5X in 500 µL cold RIPA with inhibitors. SDS 6X reducing buffer (Alfa Aesar- Thermo Fisher Scientific, Waltham, MA) was added (15 µL) and samples were incubated at 65 °C for 8 min to elute protein, after which 15 µL RIPA with inhibitors was added and protein samples were magnetically separated from beads.

### Schematics and diagrams
Schematics and diagrams were created using BioRender (Science Suite Inc.).

### Statistics
Inferential statistics were performed by null hypothesis significance testing at a significance level of 0.05 using GraphPad Prism 9. Two-tailed tests were systematically employed throughout to infer a difference against the null. Planned comparisons between two groups were analyzed by t-tests. Planned comparisons within 2-factor designs were Šidák corrected after ANOVA, except two western blot experiments run in multiple gels where t-tests were employed. Linear regressions were performed to either test for a significant association between variables (significant slope) or to compare adjusted means between two groups (significant intercept).

### Reporting summary
Further information on research design is available in the Nature Portfolio Reporting Summary linked to this article.

## Data availability
The transcriptomics data generated in this study have been deposited in the GEO repository under accession code GSE241830. Source data are provided with this paper.

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

## Acknowledgements

This work was supported by the National Institutes of Health (R01DK127444 to G.A.K.; U24DK097209 to University of Florida's Southeast Center for Integrated Metabolomics; P30NS104177 and S10OD026842 to OSU Neuroscience Imaging Core/Brown, PI), the National Institute of Food and Agriculture (NIFA-2018-67001-28246 to G.A.K.), The American Heart Association (AHA904048 to JS).

## Author contributions

J.S., J.B., I.S.B., C.M., K.R.S., K.K., S.K.M., N.N.M., V.S., G.M.L., S.J.G., and G.A.K. performed experiments; J.S., J.B., C.M., K.R.S., K.K., S.K.M., G.M.L., S.J.G., and G.A.K. analyzed data; J.S., and G.A.K. designed research studies; J.S., J.B., S.K.M., S.J.G., and G.A.K. edited the manuscript; J.S. and G.A.K. wrote the manuscript; G.A.K. conceived the project.

## Competing interests

J.S. and G.A.K. have a patent (WO 2022/120143 A1) related to this work. The remaining authors declare no competing interests.

## Inclusion and Ethics

We have complied with all relevant ethical regulations. All animal procedures were performed under the approval of The Ohio State University institutional animal care and use committee (IACUC).
