## [Peer Review File · Nature Communications]

The TAS1R2 G-protein-coupled receptor is an ambient glucose sensor in skeletal muscle that regulates NAD homeostasis and mitochondrial capacityREVIEWER COMMENTS

Reviewer #1 (Remarks to the Author):

- Fig1: Myogenin expression is repressed in the adult (Berghella et al. Genes&Development 2008, 22:2125-2138). Myogenin expression in 12-16 weeks old mice should be documented
- Fig S1b: Clarify the scale used in figure 1b.
- Paragraph 2 (page 4): TAS1R2-mediated glucose sensing regulates NAD levels coupled to PARP activity: the authors introduce the use a mouse model with genetic deletion of Tas1r2 referred to as bKO. Transcriptional and functional characterization of the mouse model is missing, there isn't any reference in material and methods to this mouse model.
- Fig S3a shows targeted nucleotide analysis in mWT and mKO performed by LC/MS. However, the reference in the text for this figure refers to bWT and bKO mice. What are bKO Tas1r2 mice?
- Fig 2e/f: The figures show NAD concentration and PAR quantification in C2C12 cells that were maintained in low glucose and subsequently enriched with glucose or 3-OMG for 6 hours. It is confusing to see that some cells treated with both Low and High Glucose conditions.
- The authors have generated several mouse models while most experiments were performed using C2C12 cells. Experiments reported in Figure 2e-g should be performed in either MuSCs or primary myoblasts.
- Fig. 2h The authors should clarify how the fasting and refeeding protocol was performed.

Reviewer #2 (Remarks to the Author):

The manuscript by Serrano and colleagues investigates the GPCR sweet taste receptor complex TAS1R2/3 as a link between extracellular glucose sensing and intracellular NAD levels in muscle. The authors demonstrate that the Tas1r2/3 genes are mildly expressed in myofibers in mice. Though there is no evidence showing endogenous protein expression, the authors demonstrated that the Tas1r2 promoter is being transcribed in muscle by using a reporter mouse model. The authors then performed generally high quality (except for western blots) and rigorous studies using knockout and transgenic mice to determine that glucose sensing (using both glucose and non-metabolisable analogues) via TAS1R2/3 regulates PARP activity to control NAD levels. The manuscript was well written, concise, and the figures were clear. I believe that this work advances the field.

Comments:

1. The manuscript is missing the expected basic phenotypic characterisation of a new mouse model with disrupted muscle glucose sensing and increased mitochondria. Minimally the authors should show body weight, tissue/organ weights, fed and fasted insulin, glucose tolerance, etc. If there is a body weight phenotype then there should be follow-up on food intake and energy expenditure. If there is a glucose tolerance phenotype then there should be mechanistic follow-up. Similar to what the authors did with the whole-body KO mouse that had better glucose tolerance.
2. The western blotting is very inconsistent and it's hard to know what to believe most of the time in Figures 2b/c/f and 4a. Just look at the spread in the quantification of the data and I don't see how you could believe any of it being judged as significant or not significant.
3. Unless I missed it, the full metabolomics should be provided as a sup file.
4. Are there more broad knock-on effects of mKO increasing NAD like decreased acetylation globally (the authors only showed PGC1a)? A small experiment to blot for total acetyl-lysine could be helpful to understand how widespread the acetylation phenotype is.
5. The authors data suggests that TAS1R2/3 is not the only mechanism linking glucose sensing to NAD because Fig 2h shows that feeding markedly decreases NAD in the mKO mice in a PARP-independent manner. The authors should discuss.
6. There's an inconsistency in rationale between finding more mitochondria in red oxidative fibres and testing endurance exercise, which is a white muscle phenotype.
7. Is it known whether PARP inhibitors alone cause these same phenotypes of increased mito mass and endurance? If not, or if it remains unknown, then statements like this won't work, "this data indicates that the increased NAD levels resulting from the inhibition of PARP activity in TAS1R2-deficient muscles (mKO) contribute to noteworthy functional adaptations," as it remains unclear if the phenotype is truly all 'resulting from the inhibition of PARP activity'. The NAD supplementation section in the discussion addresses the NAD part but not the PARP-dependent part. Please add discussion to ms.
8. The authors suggest that TAS1R2/3 inhibition could be a therapeutic approach - are there any perceived negative impacts of loss of TAS1R2/3-dependent glucose sensing? Discussion of the whole-body KO could be relevant here. Would a drug have to be targeted to muscle?
9. Minor grammar error, delete final word it in, "a human-specific TAS1R2 agonist that the mouse TAS1R2 receptor cannot sense it."

Itemized responses to reviewers' comments

NCOMMS-23-59230-T "The TAS1R2 G-protein-coupled receptor is an ambient glucose sensor in skeletal muscle that regulates NAD homeostasis and mitochondrial capacity"

We want to thank the reviewers for constructive criticism. In several instances, we have not been clear enough in our descriptions, which led to some minor misunderstandings or omissions. In this revised manuscript, we have provided further clarification and discussion of existing data to address the reviewers' suggestions and questions. Most importantly, we are including here and in the manuscript additional data based on the reviewers' suggestions. We sincerely hope the reviewers and editors find the manuscript improved and thus this revision is suitable for publication. Please find below a point-by-point response to the comments.

Revisions in the main text are highlighted in **yellow**.

REVIEWER COMMENTS

Reviewer #1 (Remarks to the Author):

"• Fig1: Myogenin expression is repressed in the adult (Berghella et al. Genes&Development 2008, 22:2125-2138). Myogenin expression in 12-16 weeks old mice should be documented"

RESPONSE: We utilized the Tg(Myog-cre)1 mouse (MGI:3530558), generously provided by Dr. Eric Olson¹, for our study. This mouse model has been carefully backcrossed and maintained in pure C57BL6/6J strain in our laboratory since 2018. The transgene in this mouse expresses cre recombinase under the regulation of the muscle-specific Myog promoter and Mef2c enhancer. Active Cre expression in MyoCre⁺ muscles is demonstrated in Figure S1b. Additionally, the figure below illustrates the specific and active recombination of Cre in skeletal muscles, while other tissues do not show such recombination.

“• Fig S1b: Clarify the scale used in figure 1b.”

RESPONSE: The scale is log10 adjusted for 18s expression (housekeeping).

“• Paragraph 2 (page 4): TAS1R2-mediated glucose sensing regulates NAD levels coupled to PARP activity: the authors introduce the use a mouse model with genetic deletion of *Tas1r2* referred to as bKO. Transcriptional and functional characterization of the mouse model is missing, there isn't any reference in material and methods to this mouse model. “

RESPONSE: We apologize for the oversight. We utilized whole-body *Tas1r2* knockout mice (bKO) (Jax#013065), which were generated and generously provided by Dr. Charles S. Zuker² (reference added in the text). This mouse model has undergone careful backcrossing and has been maintained in a pure C57BL6/6J strain in our laboratory since 2009. Both our lab and several others have extensively characterized the metabolic phenotype of this mouse model³⁻¹¹. In our study, we employed bKO mice for the initial metabolomics screen prior to generating the muscle-specific model (mKO). We have incorporated this pertinent information into the Methods section. Furthermore, this mouse model is referenced in the manuscript^{3,4,6,9,12}.

“• Fig S3a shows targeted nucleotide analysis in mWT and mKO performed by LC/MS. However, the reference in the text for this figure refers to bWT and bKO mice. What are bKO *Tas1r2* mice?”

RESPONSE: The analysis was conducted in both bWT and bKO muscles. We have rectified the typo in figure S3a and its legend. Additionally, we have discussed the bKO mice in our response above.

“• Fig 2e/f: The figures show NAD concentration and PAR quantification in C2C12 cells that were maintained in low glucose and subsequently enriched with glucose or 3-OMG for 6 hours. It is confusing to see that some cells treated with both Low and High Glucose conditions.”

RESPONSE: We apologize for any confusion caused. To clarify, cells were not treated with both Low and High glucose simultaneously. Instead, cells were initially maintained in Low glucose (as a baseline control) before being supplemented with either High glucose or High 3-OMG. The experimental design is described in the figure legend as follows: “....(e) NAD concentration and (f) Immunoblotting and quantitation of PAR in C2C12 cells that were maintained in low glucose (5mM; Glu Low) and were subsequently spiked with additional glucose (20mM; Glu High) or 3-OMG (20mM; 3-OMG High) for 6 hours. One-way ANOVA, Sidak post-hoc effect....”. We hope this clarifies the experimental procedure.

“• The authors have generated several mouse models while most experiments were performed using C2C12 cells. Experiments reported in Figure 2e-g should be performed in either MuSCs or primary myoblasts.”

RESPONSE: The experimental conditions utilized in Figure 2e,f have been previously documented in publications cited in the text^{13,14}, demonstrating that spiking C2C12 cells with high glucose leads to the suppression of NAD levels. We adopted this established model/design in C2C12 cells to facilitate a direct comparison of known NAD responses, specifically

highlighting the involvement of Tas1r2/Tas1r3 (i.e., 3-OMG and gurmarin). These *in vitro* findings are in alignment with the physiological responses observed *in vivo*, as illustrated in Figure 2h,i, and j. While we attempted to replicate similar experiments using primary cultures, we encountered significant challenges that impeded our ability to test this hypothesis. Specifically:

1. WT primary myocytes consistently exhibited high levels of NAD compared to C2C12 cells, as depicted in panel A. Additionally, the NAD levels in primary myocytes were not stable, making it difficult to establish basal steady values prior to any manipulation.

2. Shifting C2C12 cells (maintained in high glucose) to low glucose for 2 days resulted in the expected increase in NAD levels, as shown in panel B. However, primary cells did not exhibit sensitivity to ambient glucose fluctuations, as illustrated in panel C.

Due to these challenges, we were unable to utilize primary cultures to demonstrate the relationship between ambient glucose and NAD levels. To the best of our knowledge, there are no published articles demonstrating this relationship in primary myocytes.

However, in this revision, we conducted an additional experiment demonstrating that *in vivo* stimulation of TAS1R2 with aspartame leads to a decline in muscle NAD levels, eliciting a response similar to *in vivo* glucose stimulation (Fig.3c and Fig.S4f). Specifically, following an overnight fast, mWT and mTg mice were injected with saline or aspartame (total 5 injections 30 minutes apart). Aspartame induced a reduction in NAD levels in mTg mice (expressing human TAS1R2 in skeletal muscle), whereas mWT mice (incapable of sensing aspartame) remained unresponsive. These findings underscore the causative link between direct stimulation of TAS1R2 *in vivo* and muscle NAD regulation, further confirming the notion that TAS1R2 activation can modulate NAD independently from glucose metabolism.

“• Fig. 2h The authors should clarify how the fasting and refeeding protocol was performed.”

RESPONSE: The experimental description of the fasting/feeding experiment is listed under "Mouse *in vivo* treatments" section in the Methods. To enhance readability, we have now added subheadings to this section.

Reviewer #2 (Remarks to the Author):

The manuscript by Serrano and colleagues investigates the GPCR sweet taste receptor complex TAS1R2/3 as a link between extracellular glucose sensing and intracellular NAD levels in muscle. The authors demonstrate that the Tas1r2/3 genes are mildly expressed in myofibers in mice. Though there is no evidence showing endogenous protein expression, the authors demonstrated that the Tas1r2 promoter is being transcribed in muscle by using a reporter mouse model. The authors then performed generally high quality (except for western blots) and rigorous studies using knockout and transgenic mice to determine that glucose sensing (using both glucose and non-metabolisable analogues) via TAS1R2/3 regulates PARP activity to control NAD levels. The manuscript was well written, concise, and the figures were clear. I believe that this work advances the field.

Comments:

1. The manuscript is missing the expected basic phenotypic characterization of a new mouse model with disrupted muscle glucose sensing and increased mitochondria. Minimally the authors should show body weight, tissue/organ weights, fed and fasted insulin, glucose tolerance, etc. If there is a body weight phenotype then there should be follow-up on food intake and energy expenditure. If there is a glucose tolerance phenotype then there should be mechanistic follow-up. Similar to what the authors did with the whole-body KO mouse that had better glucose tolerance.

RESPONSE: We acknowledge the reviewers' request for a more comprehensive characterization of our mKO mouse model. In response, we have included a new supplemental figure (Fig. S5) presenting the following:

- a) Measurements of body mass, body length, tibia length, and liver mass to evaluate normal growth.
- b) Intraperitoneal (ip) glucose tolerance tests (ip.GTT), intra-gastric GTT (igGTT), insulin tolerance test (ITT), and ad lib plasma glucose and insulin levels.
- c) Indirect calorimetry data, including food intake and activity.
- d) Additionally, insulin responses following the overnight fast with or without refeeding are presented in Fig. S2.

Notably, no genotype effects were observed in the assessed measures.

We have observed moderate increases in muscle mass in mKO mice. However, these findings are not directly linked to the regulation of NAD and mitochondrial function, which are the primary focus of our manuscript. Over the past few years, our lab has diligently worked to elucidate the mechanisms underlying these intriguing observations. Therefore, the effects on muscle mass and their associated mechanisms are beyond the scope of the current report and will be addressed in future work.

“2. The western blotting is very inconsistent and it’s hard to know what to believe most of the time in Figures 2b/c/f and 4a. Just look at the spread in the quantification of the data and I don’t see how you could believe any of it being judged as significant or not significant.”

RESPONSE: We acknowledge the variability observed in the immunoblots for PARP1/2, PGC1a, and SIRT1. These immunoblots have been performed multiple times using different cohorts of mice, consistently yielding similar variability without a discernible genotype effect. However, it's important to note that our manuscript only makes claims about PAR levels, which consistently show a significant difference in mKO mice (approximately 40-50% lower).

In an effort to address the reviewer's concern, we conducted additional immunoblotting of these targets using samples from 7 additional mice per genotype. The results align closely with those presented in the manuscript. The figure below displays the immunoblots for the additional mice, and the quantitation includes data from all 14 mice per genotype, incorporating the new samples (total cell lysates) represented as red dots. Notably, the majority (>90%) of data points fall within ± 0.5 fold-change (as indicated by the dotted horizontal lines), with minimal differences observed between genotypes in terms of sample size effects. This approach aims to maintain full transparency and accurately represent the physiological variability of our targets.

“3. Unless I missed it, the full metabolomics should be provided as a sup file.”

RESPONSE: We conducted targeted quantitative metabolomics analysis of 30 major nucleotides. The relevant data has been compiled into a source data Excel file (xls) for inclusion with the manuscript.

4. Are there more broad knock-on effects of mKO increasing NAD like decreased acetylation globally (the authors only showed PGC1a)? A small experiment to blot for total acetyl-lysine could be helpful to understand how widespread the acetylation phenotype is.

RESPONSE: This is an important addition to our manuscript. We are including a blot showing the total acetyl-lysine profiling of mWT and mKO muscle. Notably, our analysis reveals no genotype effect on global cell acetylation, as depicted in Figure S5a.

5. The authors data suggests that TAS1R2/3 is not the only mechanism linking glucose sensing to NAD because Fig 2h shows that feeding markedly decreases NAD in the mKO mice in a PARP-independent manner. The authors should discuss.

RESPONSE: Thank you for bringing this to our attention. In the original submission, paragraph 3 of the discussion solely addresses TAS1R2-dependent and -independent effects of NAD regulation. To further emphasize this topic, in the revised manuscript, we have expanded our discussion to provide a physiological rationale for distinct mechanisms targeting NAD synthesis (TAS1R2-PAPP1 independent) and consumption (TAS1R2-PARP1 dependent) of NAD.

6. *There's an inconsistency in rationale between finding more mitochondria in red oxidative fibers and testing endurance exercise, which is a white muscle phenotype.*

RESPONSE: We appreciate the reviewer's comment. In our study, we have examined several muscles including soleus, gastrocnemius and quadriceps, and have observed similar phenotype and signaling. This suggests that the TAS1R2-mediated effects likely impact all fiber types.

Regarding the endurance protocol utilized, it's important to note that we deliberately avoided a typical ramp protocol involving increasing intensity to exhaustion. Such aggressive protocols can be influenced by mouse behavior and may not accurately reflect true endurance. Instead, we opted for a fixed moderate intensity protocol that allowed mice to run for 40-120 minutes until exhaustion. Other studies have demonstrated improved endurance through increases in NAD (NAMPT overexpression)¹⁵ or PGC1a overexpression¹⁶, both of which caused mitochondrial adaptations.

7. *Is it known whether PARP inhibitors alone cause these same phenotypes of increased mito mass and endurance? If not, or if it remains unknown, then statements like this won't work, "this data indicates that the increased NAD levels resulting from the inhibition of PARP activity in TAS1R2-deficient muscles (mKO) contribute to noteworthy functional adaptations," as it remains unclear if the phenotype is truly all 'resulting from the inhibition of PARP activity'. The NAD supplementation section in the discussion addresses the NAD part but not the PARP-dependent part. Please add discussion to ms.*

RESPONSE: Pharmacological inhibitors or genetic deficiency of PARP1 causes a very similar phenotype to our TAS1R2 deficient mice. In the Discussion section we wrote : *"..Direct inhibition of PARP1, the downstream effector of TAS1R2 signaling, also activates the NAD-SIRT1 axis leading to similar beneficial effects^{17,18}..."*. In the revised manuscript, we have included additional explicit comments about the nature of similarities between our TAS1R2-deficient mice and models involving pharmacological inhibitors or genetic deficiency of PARP1. This addition aims to provide a clearer understanding of the parallels between the different mouse models and their effects on the NAD-SIRT1 axis.

8. *The authors suggest that TAS1R2/3 inhibition could be a therapeutic approach - are there any perceived negative impacts of loss of TAS1R2/3-dependent glucose sensing? Discussion of the whole-body KO could be relevant here. Would a drug have to be targeted to muscle?*

RESPONSE: We appreciate the reviewer's inquiry. Extensive data using whole-body TAS1R2-KO (bKO) mice, although not shown in this manuscript, strongly suggest that the muscle and integrative phenotype is maintained without detectable negative effects, even in the context of obesity or aging. A separate manuscript currently in preparation will delve deeper into addressing this question.

Based on our findings, we speculate that a general inhibitor could likely exhibit efficacy for muscle-related outcomes and potentially demonstrate additional positive synergistic effects stemming from TAS1R2 inhibition in other tissues. Moreover, our laboratory has identified a partial loss-of-function variant of TAS1R2 in humans, which is associated with positive adaptations akin to those observed in mouse models of TAS1R2 deficiency.

9. *Minor grammar error, delete final word it in, "a human-specific TAS1R2 agonist that the mouse TAS1R2 receptor cannot sense it."*

RESPONSE: Thank you. We have corrected the typo.

REFERENCES

1. Li, S., Czubryt, M.P., McAnally, J., Bassel-Duby, R., Richardson, J.A., Wiebel, F.F., Nordheim, A., and Olson, E.N. (2005). Requirement for serum response factor for skeletal muscle growth and maturation revealed by tissue-specific gene deletion in mice. *Proceedings of the National Academy of Sciences of the United States of America* *102*, 1082-1087. 10.1073/pnas.0409103102.
2. Zhao, G.Q., Zhang, Y., Hoon, M.A., Chandrashekar, J., Erlenbach, I., Ryba, N.J., and Zuker, C.S. (2003). The receptors for mammalian sweet and umami taste. *Cell* *115*, 255-266.
3. Kyriazis, G.A., Soundarapandian, M.M., and Tyrberg, B. (2012). Sweet taste receptor signaling in beta cells mediates fructose-induced potentiation of glucose-stimulated insulin secretion. *Proc.Natl.Acad.Sci.U.S.A* *109*, E524-E532. 1115183109 [pii];10.1073/pnas.1115183109 [doi].
4. Kyriazis, G.A., Smith, K.R., Tyrberg, B., Hussain, T., and Pratley, R.E. (2014). Sweet taste receptors regulate basal insulin secretion and contribute to compensatory insulin hypersecretion during the development of diabetes in male mice. *Endocrinology* *155*, 2112-2121. 10.1210/en.2013-2015.
5. Smith, K.R., Hussain, T., Karimian Azari, E., Steiner, J.L., Ayala, J.E., Pratley, R.E., and Kyriazis, G.A. (2016). Disruption of the sugar-sensing receptor T1R2 attenuates metabolic derangements associated with diet-induced obesity. *American journal of physiology. Endocrinology and metabolism* *310*, E688-E698. 10.1152/ajpendo.00484.2015.
6. Smith, K., Karimian Azari, E., LaMoia, T.E., Hussain, T., Vargova, V., Karolyi, K., Veldhuis, P.P., Arnoletti, J.P., de la Fuente, S.G., Pratley, R.E., et al. (2018). T1R2 receptor-mediated glucose sensing in the upper intestine potentiates glucose absorption through activation of local regulatory pathways. *Mol Metab* *17*, 98-111. 10.1016/j.molmet.2018.08.009.
7. Serrano, J., Smith, K.R., Crouch, A.L., Sharma, V., Yi, F., Vargova, V., LaMoia, T.E., Dupont, L.M., Serna, V., Tang, F., et al. (2021). High-dose saccharin supplementation does not induce gut microbiota changes or glucose intolerance in healthy humans and mice. *Microbiome* *9*, 11. 10.1186/s40168-020-00976-w.
8. Serrano, J., Seflova, J., Park, J., Pribadi, M., Sanematsu, K., Shigemura, N., Serna, V., Yi, F., Mari, A., Procko, E., et al. (2021). The Ile191Val is a partial loss-of-function variant of the TAS1R2 sweet-taste receptor and is associated with reduced glucose excursions in humans. *Mol Metab* *54*, 101339. 10.1016/j.molmet.2021.101339.
9. Serrano, J., Meshram, N.N., Soundarapandian, M.M., Smith, K.R., Mason, C., Brown, I.S., Tyrberg, B., and Kyriazis, G.A. (2022). Saccharin Stimulates Insulin Secretion

- Dependent on Sweet Taste Receptor-Induced Activation of PLC Signaling Axis. *Biomedicines* *10*. 10.3390/biomedicines10010120.
10. Simon, B.R., Parlee, S.D., Learman, B.S., Mori, H., Scheller, E.L., Cawthorn, W.P., Ning, X., Gallagher, K., Tyrberg, B., Assadi-Porter, F.M., et al. (2013). Artificial sweeteners stimulate adipogenesis and suppress lipolysis independently of sweet taste receptors. *The Journal of biological chemistry* *288*, 32475-32489. 10.1074/jbc.M113.514034.
 11. Simon, B.R., Learman, B.S., Parlee, S.D., Scheller, E.L., Mori, H., Cawthorn, W.P., Ning, X., Krishnan, V., Ma, Y.L., Tyrberg, B., and MacDougald, O.A. (2014). Sweet taste receptor deficient mice have decreased adiposity and increased bone mass. *PloS one* *9*, e86454. 10.1371/journal.pone.0086454.
 12. Karimian Azari, E., Smith, K.R., Yi, F., Osborne, T.F., Bizzotto, R., Mari, A., Pratley, R.E., and Kyriazis, G.A. (2017). Inhibition of sweet chemosensory receptors alters insulin responses during glucose ingestion in healthy adults: a randomized crossover interventional study. *The American journal of clinical nutrition* *105*, 1001-1009. 10.3945/ajcn.116.146001.
 13. Canto, C., Jiang, L.Q., Deshmukh, A.S., Matakı, C., Coste, A., Lagouge, M., Zierath, J.R., and Auwerx, J. (2010). Interdependence of AMPK and SIRT1 for metabolic adaptation to fasting and exercise in skeletal muscle. *Cell Metab* *11*, 213-219. 10.1016/j.cmet.2010.02.006.
 14. Frederick, D.W., Davis, J.G., Davila, A., Jr., Agarwal, B., Michan, S., Puchowicz, M.A., Nakamaru-Ogiso, E., and Baur, J.A. (2015). Increasing NAD synthesis in muscle via nicotinamide phosphoribosyltransferase is not sufficient to promote oxidative metabolism. *The Journal of biological chemistry* *290*, 1546-1558. 10.1074/jbc.M114.579565.
 15. Costford, S.R., Brouwers, B., Hopf, M.E., Sparks, L.M., Dispagna, M., Gomes, A.P., Cornell, H.H., Petucci, C., Phelan, P., Xie, H., et al. (2018). Skeletal muscle overexpression of nicotinamide phosphoribosyl transferase in mice coupled with voluntary exercise augments exercise endurance. *Mol Metab* *7*, 1-11. 10.1016/j.molmet.2017.10.012.
 16. Lee, S., Leone, T.C., Rogosa, L., Rumsey, J., Ayala, J., Coen, P.M., Fitts, R.H., Vega, R.B., and Kelly, D.P. (2017). Skeletal muscle PGC-1beta signaling is sufficient to drive an endurance exercise phenotype and to counteract components of detraining in mice. *American journal of physiology. Endocrinology and metabolism* *312*, E394-E406. 10.1152/ajpendo.00380.2016.
 17. Bai, P., Canto, C., Oudart, H., Brunyanszki, A., Cen, Y., Thomas, C., Yamamoto, H., Huber, A., Kiss, B., Houtkooper, R.H., et al. (2011). PARP-1 inhibition increases mitochondrial metabolism through SIRT1 activation. *Cell Metab* *13*, 461-468. 10.1016/j.cmet.2011.03.004.
 18. Pirinen, E., Canto, C., Jo, Y.S., Morato, L., Zhang, H., Menzies, K.J., Williams, E.G., Mouchiroud, L., Moullan, N., Hagberg, C., et al. (2014). Pharmacological Inhibition of poly(ADP-ribose) polymerases improves fitness and mitochondrial function in skeletal muscle. *Cell Metab* *19*, 1034-1041. 10.1016/j.cmet.2014.04.002.

REVIEWERS' COMMENTS

Reviewer #1 (Remarks to the Author):

The inability to reproduce the results in C2C12 cells in primary myoblasts or MuSCs (changes in NAD after glucose stimulation) raise some concerns on the physiological meaning of the experiments performed in C2C12 cells.

Reviewer #2 (Remarks to the Author):

No further comments